# Preliminary Results of the FASM Study, an On-Going Italian Active Pharmacovigilance Project

**DOI:** 10.3390/ph13120466

**Published:** 2020-12-15

**Authors:** Giorgia Teresa Maniscalco, Vincenzo Brescia Morra, Ciro Florio, Giacomo Lus, Gioacchino Tedeschi, Maddalena Cianfrani, Renato Docimo, Stefania Miniello, Felice Romano, Leonardo Sinisi, Daniele L. A. Spitaleri, Giuseppe Longo, Ugo Trama, Maria Triassi, Cristina Scavone, Annalisa Capuano

**Affiliations:** 1Multiple Sclerosis Center, “A. Cardarelli” Hospital, 80131 Naples, Italy; gtmaniscalco@libero.it (G.T.M.); mamifaccilpiacere@tin.it (C.F.); 2Neurological Clinic and Stroke Unit, “A. Cardarelli” Hospital, 80131 Naples, Italy; 3MS Clinical Care and Research Centre, Department of Neuroscience, “Federico II’’ University, 80131 Naples, Italy; vincenzo.bresciamorra2@unina.it; 4Department of Medical, Surgical, Neurological, Metabolic and Aging Sciences, II Clinic of Neurology University of Campania “Luigi Vanvitelli”, 80131 Naples, Italy; giacomo.lus@unicampania.it; 5I Clinica Neurologica e Neurofisiopatologia Azienda Ospedaliera Universitaria—Università della Campania “Luigi Vanvitelli”, 80138 Naples, Italy; gioacchino.tedeschi@unicampania.it; 6AOU San Giovanni di Dio Ruggi D’aragona, 84131 Salerno, Italy; maddalena.cianfrani@gmail.com; 7MS Center, “San Giuseppe Moscati” Hospital, 81031 Aversa, Italy; renato.docimo@gmail.com; 8Neurological Outpatient Service, “San Giuseppe Moscati” Hospital, 81031 Aversa, Italy; 9Neurology and Stroke Unit, “Sant’Anna e San Sebastiano” Hospital, 81100 Caserta, Italy; stefania.miniello@virgilio.it; 10Multiple Sclerosis Center, CTO Hospital, AORN “Ospedali dei Colli”, 80131 Naples, Italy; felrom@inwind.it; 11Neurological and Stroke Unit, CTO Hospital, AORN “Ospedali dei Colli”, 80131 Naples, Italy; 10Neurological Unit and MS Center, San Paolo Hospital, ASL Napoli 1 Centro, 80125 Naples, Italy; leosinisi@libero.it; 13MS Center, San Paolo Hospital, ASL Napoli 1 Centro, 80125 Naples, Italy; 14UOC Neurologia-Centro Sclerosi Multipla-AORN San G. Moscati, 83100 Avellino, Italy; daspital@tin.it; 15Direzione Generale, “A. Cardarelli” Hospital, 80131 Naples, Italy; direzione.generale@aocardarelli.it; 16Regional Pharmaceutical Unit, Campania Region, 80143 Naples, Italy; ugo.trama@regione.campania.it; 17Department of Public Health, “Federico II’’ University, 80138 Naples, Italy; triassi@unina.it; 18Department of Experimental Medicine, University of Campania “Luigi Vanvitelli”, 80138 Naples, Italy; annalisa.capuano@unicampania.it; 19Regional Center of Pharmacovigilance and Pharmacoepidemiology of Campania Region, 80138 Naples, Italy

**Keywords:** multiple sclerosis, disease-modifying therapies, safety, adverse drugs reactions, FASM study, pharmacovigilance project

## Abstract

Background and aim: Disease-modifying therapies (DMTs) used in multiple sclerosis (MS) have distinct safety profiles. In this paper, we report preliminary results of an on-going pharmacovigilance project (the FASM study). Results: Neurologists working at involved multiple sclerosis centers collected 272 Individual Case Safety Reports (ICSRs). Adverse drug reactions (ADRs) mainly occurred in adult patients and in a higher percentage of women compared to men. No difference was found in ADRs distribution by seriousness. The outcome was reported as favorable in 61% of ICSRs. Out of 272 ICSRs, almost 53% reported dimethyl fumarate, fingolimod and IFN beta 1a as suspected. These medications were commonly associated to the occurrence of ADRs related hematological, gastrointestinal, general, infective or cancer disorders. The median time to event (days) was 177 for dimethyl fumarate, 1058 for fingolimod and 413 for IFN beta 1a. The median time to event for the remaining suspected drugs was 226. Conclusion: We believe that our results, together with those that will be presented at the end of the study, may bring new knowledge concerning the safety profile of DMTs and their proper use. This will provide the opportunity to draw new recommendations both for neurologists and patients.

## 1. Introduction

Multiple sclerosis (MS) is an inflammatory-mediated demyelinating disease of the Central Nervous System (CNS) caused by an immune dysregulation associated with genetic and environmental factors, such as Epstein–Barr virus infection and cigarette smoking [1,2], while the potential role of Vitamin D in neurological diseases, including MS, is mostly unclear and still controversial [3,4,5,6]. MS approximately affects 2.5 million people worldwide. The disease generally occurs in people aged 20–40 years, being more common in women than in men [7]. Even though relapsing-remitting MS (RR-MS) is the most common MS form, most patients gradually develop into a chronic progressive phase, progressive MS (PMS). In order to prevent and reduce the number of relapses and delay the progression of the disease, disease-modifying therapies (DMTs) are prescribed [8]. In this context, the pipeline of MS therapeutic options has significantly improved thanks to the introduction of many innovative drugs. Indeed, following the approval in Europe of injectable interferons (IFNs) in the 1990s, new medications, some of which biological agents, obtained the marketing authorization, including glatiramer acetate (authorized by the EMA in 2001), natalizumab (2006), fingolimod (2011), alemtuzumab and teriflunomide (2013), dimethyl fumarate (2014), cladribine and ocrelizumab (2017) [9]. Among those medicines, IFNsβ, glatiramer acetate, dimethyl fumarate and teriflunomide represent first-line treatments, while the remaining drugs are prescribed in case of unsatisfactory response or when previous drugs are not well tolerated or in case of very active MS. DMTs have distinct pharmacodynamics properties, resulting in immunomodulatory and anti-inflammatory response, and consequently individual efficacy and tolerability profiles. Indeed, first-line drugs usually show a moderate efficacy and high safety profiles, while second-line agents are associated to higher efficacy with increased safety risks. Clinical trials that have compared IFNβ and glatiramer acetate showed that the efficacy of these medications is similar, while common adverse drug reactions (ADRs) are represented by flu-like symptoms and injection site reactions [10,11]. Even though serious ADRs seem to be rare, few safety issues have emerged for some of the recently authorized drugs. For instance, fingolimod was associated with an increased risk of infections, cancers, and hepatotoxicity. The risk of progressive multifocal leukoencephalopathy (PML) was rarely reported with fingolimod, while it was significantly higher for natalizumab [2,12]. Ocrelizumab, a monoclonal antibody targeting B-cell, was mainly associated with infusion-related reactions, while daclizumab was related to the occurrence of hepatotoxicity and fatal immune reactions affecting the brain, liver and other organs [2]. For this reason, daclizumab has been retracted from the market. Furthermore, the EMA recently recommended the restriction of the use for alemtuzumab due to reports of serious infections, cardiovascular and immune-related disorders, including deaths [13].

Considering that several DMTs were recently approved, pharmacovigilance activities represent a fundamental tool for the collection of new safety data, improving the knowledge on the benefit–risk ratio of these drugs. Taking this into account, we report the preliminary results of a pharmacovigilance study performed on Individual Case Safety Reports (ICSRs) collected between 1 September 2018 and 31 December 2019 in 10 MS centers of the Campania Region (Southern Italy) (the study “Farmacovigilanza Attiva Sclerosi Multipla”—FASM) through the Italian Pharmacovigilance database (Rete Nazionale di Farmacovigilanza—RNF) that is managed by the Italian Medicine Agency (AIFA). We aim to present a descriptive analysis of safety data collected during 16 months of activities and to describe the main characteristics of ICSRs, with a focus on the three most commonly reported suspected DMTs.

## 2. Results

### 2.1. Overall Results

During the first 16 months of the FASM project, neurologists working at involved MS Centers collected 272 ICSRs that reported as suspected a DMT used in the treatment of MS.

ADRs mainly occurred in adult patients (median age: 42 years; IQR: 30–50) and in a higher percentage of women compared to men (67.7% vs. 31.6%). No substantial difference was found in ADRs distribution by seriousness (50.7% not serious vs. 49.3% serious). The outcome was reported as favorable in 61% of ICSRs and as unfavorable in 21% of ICSRs (in 18% of ICSRs the outcome was not reported). In 99% of ICSRs (*n* = 269) no suspected drugs other than those indicated for the treatment of MS were reported; in 31.6% of ICSRs concomitant medications were reported (Table 1). Among ICSRs reporting serious ADRs, 3 were defined as “serious—death” and 3 as “serious—life-threat”. Specifically, among ICSRs reporting ADRs that led to patient’s death, cases No. 1 and 2 reported a preterm birth occurred in pregnant women taking natalizumab and glatiramer acetate, respectively. In case No. 2, methylprednisolone was reported as suspected drug, together with glatiramer acetate. The remaining fatal case (case No. 3) concerned a male patient who experienced respiratory distress after taking ocrelizumab. For all fatal cases the Naranjo algorithm determined causality as possible (Table 2). ICSRs reporting ADRs that were considered as “serious—life-threat” referred to cases of meningitis and encephalitis after alemtuzumab and fingolimod therapy. As for fatal cases, the Naranjo algorithm determined causality as possible too (Table 2). 

Looking at each suspected DMT some differences in previously described characteristics were noted (Table 1). For instance, with regard to the median age, patients experiencing ADRs following alemtuzumab, cladribine and natalizumab were younger (median age: 39, 27 and 31, respectively) compared with patients experiencing ADRs following the administration of other DMTs. We also found that the percentage of female gender was higher compared to that reported for all ICSRs for the following suspected drugs: alemtuzumab (82%), cladribine (71%), dimethyl fumarate (73%), IFN beta1a (76%), natalizumab (71%), pegIFN beta 1a (77%) and teriflunomide (74%). We also found that the percentage of serious ADRs was higher to that reported for all ICSRs for the following suspected drug: alemtuzumab (73%), cladribine (57%), fingolimod (58%) and natalizumab (62%). Lastly, we found that the percentage of favorable outcomes was higher to that reported for all ICSRs for the following suspected drugs: glatiramer acetate (77.8%), IFN beta1a (70.7%), IFN beta1b (100%), ocrelizumab (72%), pegIFN beta 1a (77%), teriflunomide (68%). On the contrary, alemtuzumab-induced ADRs showed more commonly an unfavorable outcome (45%) (Table 1). 

### 2.2. Focus on Drugs Most Commonly Reported as Suspected

Out of 272 ICSRs, almost 53% (*n* = 144) reported dimethyl fumarate (*n* = 41), fingolimod (*n* = 62) and IFN beta 1a (*n* = 41) as suspected (Table 3). Because in each ICSR more than one ADR could be reported, we observed a total of 187 ADRs (53 for dimethyl fumarate, 80 for fingolimod and 54 for IFN beta 1a). SOCs that were most commonly reported for these medications were “Blood and lymphatic system disorders” (32.1% of all dimethyl fumarate-induced ADRs and 20% of all IFN beta 1a-induced ADRs), “General disorders and administration site conditions” (44% of all IFN beta 1a -induced ADRs), “Infections and infestations” (18.75% of all fingolimod-induced ADRs), “Neoplasms benign, malignant and unspecified” (21.25% of all fingolimod-induced ADRs) and “Gastrointestinal disorders” (20.8% of all dimethyl fumarate-induced ADRs). Regarding to the distribution by PT, we found that: for dimethyl fumarate the most reported ADRs were lymphocytopenia for the SOC “Blood and lymphatic system disorders” (12/17) and heartburn for the SOC “Gastrointestinal disorders” (7/11); for fingolimod the most reported ADRs were Herpes zoster infection and cystitis for the SOC “Infections and infestations” (9/15) and breast cancer and melanoma for the SOC “Neoplasms benign, malignant and unspecified” (6/17); for IFN beta 1a the most reported ADRs were leucopenia for the SOC “Blood and lymphatic system disorders” (5/11) and flu-like symptoms for the SOC “General disorders and administration site conditions” (16/24) (Table 3).

The TTE was computed using the date of ADR occurrence and the date of starting therapy. For 6 ICSRs (2 for each drug) these dates were not available and therefore the TTE was not computed. The median TTE (days) was 177 for dimethyl fumarate (range: 0–1689), 1058 for fingolimod (range: 0–3862) and 413 for IFN beta 1a (range: 0–7066). The median TTE for the remaining suspected drugs was 226 (range: 0–4424) (Figure 1). Stratifying TTE values by SOCs further differences were highlighted (Table 4). For instance, ADRs belonging to the SOC “Blood and lymphatic system disorders” tended to appear earlier with dimethyl fumarate compared to fingolimod and IFN beta1a (median TTE: 224, 1227 and 1518 days, respectively). The same was observed for dimethyl fumarate-induced ADRs belonging to the SOC “Infections and infestations” that occurred after a median TTE of 943 days compared to 1486 days with fingolimod. We also observed that ADRs belonging to the SOC “Neoplasms benign, malignant and unspecified” occurred earlier with fingolimod compared to IFN beta1a (median TTE: 1589 days vs. 3681 days, respectively). Lastly, ADRs belonging to the SOC “Skin and subcutaneous tissue disorders” were delayed with dimethyl fumarate compared to IFN beta 1a (679 days vs. 72 days, respectively) (Table 4).

## 3. Discussion

### 3.1. Overall Results

We have presented the preliminary results of the FASM project, an on-going multicenter Italian active pharmacovigilance study started in September 2018 that aims to analyze the safety profile of DMTs used in a real life setting of Southern Italy for the treatment of MS. For this purpose, participating neurologists started to collect data on ADRs occurred in their patients, during their routine clinical practice and by filling in the ICSR approved by the AIFA. Some of participating neurologists have already experience in collecting pharmacovigilance data [14,15,16,17,18,19]. 

During the first 16 months of activity, 272 ICSRs were collected and sent to the RNF in the Campania region. ADRs mainly occurred in adult patients and in a slightly higher percentage of females. For some MS drugs, including alemtuzumab, cladribine, dimethyl fumarate, IFN beta1a, natalizumab, PegIFN beta1a and teriflunomide, the percentage of female patients who experienced ADRs was higher compared to the overall data. In our opinion, these data are not surprising if we consider that MS shows the highest prevalence in the age group 35–64 years and that a female predominance is observed [20], including in Italy [21]. Indeed, the prevalence ratio of multiple sclerosis of women to men is approximately 2.3–3.5:1 [22] and women seem to experience ADRs nearly twice as often as men [23]. As a matter of fact, in previous pharmacovigilance studies, independently by the suspected drug, female patients seemed to be more prone to experience ADRs due to changes in pharmacokinetic/pharmacodynamic behaviors of drugs and to hormonal factors as well [24,25,26]. 

With regard to ADRs distribution by seriousness, even though for all ICSRs we did not find substantial difference (50.7% of ICSRs reported not serious ADRs vs. 49.3% of ICSRs reported serious ADRs), some differences were highlighted looking at the single suspected DMT. For instance, serious ADRs seemed to be more common with alemtuzumab, cladribine, fingolimod and natalizumab and, among these medications, three were associated to fatal or life threating ADRs. Furthermore, alemtuzumab-induced ADRs were associated to worse outcomes too. Even though the number of ICSRs associated with each single DMT is quite low, considerations need to be made. First and foremost, alemtuzumab and natalizumab are monoclonal antibodies and it is well known that these drugs are more commonly associated with serious ADRs [27,28,29]. Second, literature data suggest that these medications may lead to the occurrence of serious ADRs, which can be fatal, that may include opportunistic infections, tumors, infusion-related events and adverse effects on pregnancy outcomes [30]. Indeed, in our study, natalizumab was associated to a fatal case of preterm birth. In this regard, it is well known that the use of MS medications during pregnancy still represents a matter of concern since there are no drugs that can be defined completely safe for this frail population; therefore, in order to avoid possible risks of exposing the unborn fetus to DMTs, pregnant women often discontinue their pharmacological treatment increasing the risk of relapses and disease progression [31]. The results of a retrospective study of chart review of 15 births from mothers receiving natalizumab revealed that complications during the second and third trimester of pregnancy were quite common but no effect on mortality or morbidity were noted [32]. A further study reported that the use of natalizumab, especially during the third trimester, is associated with higher risk of thrombocytopenia and anemia in the newborns [33]. The effects of natalizumab on pregnancy outcomes may originate from the same mechanism of action of the drug, the inhibition of α4 integrins, which seems to affect processes of fertilization, placental development, embryo implantation, hematopoiesis, and cardiac development [34]. In addition, it should be underline that according to suggestions from a recent UK consensus, the last dose of natalizumab during pregnancy is recommended at 34 weeks, while the restart of the therapy is recommended soon after the birth. This is due to two main reasons: first, women taking natalizumab are those with severe and active disease and, second, natalizumab is not able to cross the placenta during the first trimester, while it is actively transported during the second and third trimesters. Lastly, the summaries of product characteristic for glatiramer, IFN-B, dimethyl fumarate and natalizumab state that these treatments should be used in pregnancy only if the benefits outweigh the risks [35].

We also observed two life-threating ADRs (a case of bacterial meningitis and pneumonitis and a case of autoimmune encephalitis) induced by alemtuzumab. Of note, both ADRs are already mentioned, together with hemolytic anemia, acute coronary syndrome, pneumonitis, PML and Lambert–Eaton myasthenia, among the rare but serious alemtuzumab-induced ADRs [36]. Indeed, the risk of infections is increased during alemtuzumab treatment since the drug targets CD8+ and CD4+-T-cells, which are involved in T-cell mediated bacterial clearance [37]. On the other hand, alemtuzumab can be also associated to autoimmune disorders, mainly as a result of an exaggerated B-cell recovery in the absence of T-cell regulation that leads to antibody-mediated B-cell autoimmunity [38,39]. In this respect, Cossburn et al. reported that the cumulative risk for the development of autoimmune diseases following alemtuzumab is 22.2% [40]. A possible explanation for the higher percentage of serious ADRs for fingolimod could be found in the type of toxicity this drug is related to. Indeed, a post-marketing analysis of 54,000 patient years showed that fingolimod was related to the occurrence of serious infections, while other studies showed an increased rate of basal cell carcinomas and melanomas and immune-mediated conditions as well [2,12,41,42,43]. Lastly, we found one fatal case of respiratory distress in a patient treated with ocrelizumab. This event was previously observed and it is currently reported in the product monograph [44]. 

On the other hand, in our study ADRs were more frequently reported as not serious with dimethyl fumarate, glatiramer acetate, IFN beta1a and beta1b, ocrelizumab and pegIFN beta 1a; furthermore, all those drugs, except for dimethyl fumarate, were more commonly related to favorable outcomes. Among these medications, dimethyl fumarate has demonstrated a good tolerability profile in phase 3 clinical trials, being associated to ADRs, such as flushing, gastrointestinal events and mild infections, including nasopharyngitis, urinary tract infections, upper respiratory tract infection, bronchitis and influenza, while no case of malignancies have been reported [45,46,47]. Similarly, literature data suggested that glatiramer acetate is not associated with serious ADRs usually observed with the newer MS therapies (i.e., infections, malignancy or autoimmune disorders) [48]. Surprisingly, glatiramer acetate was associated to a case of preterm birth in our study, even though literature data suggested that it might represent one of the safest drugs to be used during pregnancy, since it seems to not affect fertility, pregnancy or fetal outcomes [49]. Data from a phase III clinical trial involving IFN revealed that serious ADRs were rarely reported and that were mainly represented by cases of depression, suicide attempt, MS aggravation and dystonia, but no one of these cases was classified as definite interferon beta 1a side effect [50]. Overall, it should be noted that in our study more than 30% of ICSRs reported concomitant medications, whose role in the occurrence of ADRs could not be excluded.

### 3.2. Dimethyl Fumarate, Fingolimod and IFN Beta1a-Related ICSRs

Out of 272 ICSRs sent to the RNF in the Campania region, 144 (covering 187 ADRs) reported dimethyl fumarate, fingolimod or IFN beta 1a as suspected drugs. In our opinion, the higher number of ICSRs reporting ADRs related to these drugs doesn’t mean that they are less safe, but merely that they are more used than others. Indeed, literature reported that IFN beta 1a and glatiramer acetate are widely used as first line treatments due to their moderate efficacy and favorable long-term safety profile [51]. Furthermore, fingolimod is approved as a second-line treatment in the European Union but as a first-line treatment in the United States [52]. In line with our results, the results of a study carried out in another Italian region (Veneto), which aimed to describe the trend in DMTs utilization and persistence to treatment in 3025 MS patients, reported that dimethyl fumarate and fingolimod were the most commonly prescribed drugs [53].

In our study, dimethyl fumarate-induced ADRs were predominantly related to the SOCs “Blood and lymphatic system disorders” (*n* = 17; median TTE: 224 days), mainly represented by cases of lymphocytopenia, and “Gastrointestinal disorders” (*n* = 11; median TTE: 92 days), mainly represented by cases of heartburn. Even though lymphopenia and leukopenia are currently identified as uncommon ADRs associated with dimethyl fumarate [54], a recent retrospective study of 194 RRMS patients treated with this drug at the Beth Israel Deaconess Medical Center revealed that 38% of patients developed lymphopenia [55]. Since lymphocytopenia represents a risk factor for opportunistic infections, blood tests every 6–8 weeks is recommended [56]. Regarding to the TTE, a recent single-center study revealed that a total of 11 RRMS patients experienced grade 3 lymphopenia after a mean of 501.9 days of treatment (range 172–1064; four patients developed lymphopenia within the first year of treatment, 5 within the second year and 2 after the second year) [57]. In line with our data, literature suggests that patients receiving dimethyl fumarate often experience gastrointestinal ADRs, such as diarrhea, nausea, abdominal pain, vomiting, and dyspepsia and that those ADRs occur most frequently within the first 10–12 weeks of treatment initiation [58,59].

In our study, fingolimod-induced ADRs were mainly related to the SOCs “Neoplasms benign, malignant and unspecified” (*n* = 17; median TTE: 1589.5) that include cases of breast cancer, melanoma and melanocytic nevus and “Infections and infestations” (*n* = 15; median TTE: 1486.5), mainly represented by herpes zoster infection and cystitis. In line with our findings, safety data from FREEDOMS and TRANSFORMS trials confirmed that fingolimod is commonly associated with infections (including PML, varicella-zoster-virus and herpes-simplex-virus infections), hematological toxicity and increase in hepatic enzymes. A further study highlighted increased rates of cutaneous malignancies [60,61,62]. Furthermore, the results of a recent Italian pharmacovigilance study showed that fingolimod was associated with the occurrence of infections, including one case of candidiasis, one case of influenza, and one case of urinary tract infection, and hematological toxicity, including one case of leukopenia [8]. Although MS drugs may increase the susceptibility to infections, given their immune-suppressive/modulating mechanism of action, it should be underline that MS patients are exposed themselves to an increased risk of infection from communicable diseases, which may lead to severe disease relapses [63]. Therefore, all patients should be strictly monitored in order to prevent that risk. As expected, in our study cancer cases were reported after a longer TTE. In line with this, some case reports described the occurrence of cancer in MS patients at least after 4 years the starting of therapy [64,65,66].

Lastly, in our study, IFN-induced ADRs were mainly related to the SOCs “Blood and lymphatic system disorders” (*n* = 11; median TTE: 1518) that include cases of leucopenia and neutropenia and “General disorders and administration site conditions” (*n* = 24; median TTE: 188), mainly represented by flu-like symptoms. According to literature data, IFNs can be commonly associated to the occurrence of hematological, general and systemic toxicities that include flu-like symptoms, injection-site reactions, leukopenia and lymphopenia [67,68]. In particular, the TTE for these ADRs ranges from 1 week [69] to 3–6 months [70].

## 4. Materials and Methods 

### 4.1. FASM Project

This is an ongoing active pharmacovigilance study on the safety of DMTs used in patients affected by MS, started in September 2018 among ten hospitals and/or Institutes for Treatment and Research located in the Campania Region (South of Italy), which covers a population approximately equal to 5.000 MS patients. For each hospital and/or Institutes for Treatment and Research participating in the study, one neurologist and one or two collaborators collected data on ADRs occurring in patients receiving a DMT.

The primary aim of the FASM project is to analyze all ADRs related to IFN beta-1a/1b, pegIFN beta-1a, glatiramer acetate, teriflunomide, fingolimod, cladribine, dimethyl fumarate, alemtuzumab, ocrelizumab or natalizumab, that were identified during the routine clinical practice. Secondary aims are: identify preventable ADRs, evaluate the effects of the pharmacovigilance project on the reporting of ADRs (in terms of number of collected ICSRs during the project compared to that collected before the study) and improve the exchange of information relating to the management of the main ADRs among clinicians of each participating MS Center. In this paper we present preliminary results related to the primary aim. Results related to secondary aims will be presented at the end of the study.

#### 4.1.1. Collection of Individual Case Safety Reports

During their routine clinical activities, neurologists who decided to participate to the FASM project started to collect, since September 2018, data on ADRs occurring in patients receiving a DMT, filling in the ICSR approved by the AIFA. Once collected, ICSRs are sent to each respective Responsible person of Pharmacovigilance, who proceed with their upload into the RNF. This database was established by the AIFA in 2001. According to Italian pharmacovigilance rules, each Italian region shall carry out post-marketing surveillance activities, including those related to the collection and analysis of ICSRs, through a regional Center of Pharmacovigilance. The Campania regional Center of Pharmacovigilance was activated with the implementation of Legislative Decree No. 95 of 2003 and the Resolution No. 2530 of 6 August 2003 [26].

#### 4.1.2. Data Analysis

ICSRs received by the Campania regional Center of Pharmacovigilance from September 2018 to December 2019 that reported IFN beta-1a/1b, pegIFN beta-1a, glatiramer acetate, teriflunomide, fingolimod, cladribine, dimethyl fumarate, alemtuzumab, ocrelizumab, natalizumab as suspected were evaluated. 

We performed a descriptive analysis of all ICSRs stratifying by median age (IQR), sex, seriousness, outcome, suspected DMT(s), number of suspected drug(s) other than those indicated for the treatment of MS and number of concomitant medications. As described in the ICH-E2A (Good Pharmacovigilance Practices Annex IV), the seriousness of ADRs was categorized as: serious—death; serious—hospitalization or its prolongation; serious—persistent or significant disability or incapacity; serious—life-threat; serious—congenital anomaly/birth defect; serious—clinically relevant; not serious; not defined. The outcome was categorized as favorable (completely resolved or improved) or unfavorable (resolved with sequelae, unchanged or death) [71].

Once identified the 3 most common suspected DMTs, we performed a descriptive analysis stratifying by System Organ Class (SOC), Preferred Term (PT) and median Time to Event (TTE). 

The Italian Regional Centers of Pharmacovigilance use the Naranjo algorithm in order to establish the strength of the relationship between a suspected drug and ADR(s). Therefore, as part of the routine pharmacovigilance activities of the Campania regional Center of Pharmacovigilance, the Naranjo algorithm was applied for all ICSRs evaluated in this study. All scores ranged between possible and certain reports were considered reasonable for causality. Given the clinical impact of serious ADRs that were life threating or resulted in death, we decided to show Naranjo algorithm results exclusively for these cases.

Descriptive statistics was used to summarize data. Categorical data were reported as frequencies and percentages, whereas continuous data were reported as median (IQR). Data management was carried out using Excel program. 

#### 4.1.3. Compliance with Ethical Standard

All procedures and experimental protocols requested for the collection of ICSRs were carried out according to rules established by the AIFA and all methods were carried out in accordance with relevant pharmacovigilance guidelines and regulations.

Safety data deriving from the Italian spontaneous reporting system are anonymous and in compliance with the ethical standard. According to the rules on pharmacovigilance as set out in Directive 2001/83/EC and Regulation (EC) No 726/2004 and the Commission Implementing Regulation (EU) No 520/2012 of 19 June 2012, which cover the examination of ICSRs and aggregated data from active surveillance systems, the approval of ethic committee is waived for pharmacovigilance studies, like this presented in this manuscript. Therefore, no further ethical measures neither patient’s consent to participate was required. 

## 5. Study Strengths and Limitations

It is well known that post-marketing surveillance activities have several limitations and among the major for the spontaneous reporting system the so-called underreporting can be found. In this regard, it should be highlighted that healthcare professionals (HCPs) seem to be especially prone to report more serious ADRs that result in hospitalization, are life threatening or result in death [65]. Despite this is an active pharmacovigilance project that is aimed to improve the reporting from HCPs, this factor may have induced a sort of underreporting in our study. For instance, neurologists are aware that fingolimod-induced leukopenia is almost immediate but probably they have reported only serious cases that had longer TTE. Furthermore, it should be recall that spontaneous reporting systems collect all ICSRs that report a “suspected” ADR. Therefore, even though we have applied the causality assessment through the Naranjo algorithm, there is no certainty about the causal association of drug/ADR. In addition, data collected from spontaneous reporting systems could be incomplete and incorrect. For instance, the lack of specific information (such as that related to the ADR’s outcome) or an error in reporting MS treatment interval and/or date of ADR occurrence (both used for the evaluation of the causality assessment and the computation of the TTE) or in concomitant medications or diseases (that may have contributed to the occurrence of the ADR) cannot be ruled out. In addition, at the moment, we reported only the preliminary results of this ongoing study. Therefore, we have analyzed a limited number of ICSRs. For instance, for some suspected drugs, such as ocrelizumab, PegIFN beta 1a and teriflunomide, less than 20 ICSRs were reported. Therefore, the proper evaluation of those ICSRs is inevitably affected by intrinsic limitations.

Nevertheless, we were able to observe the use of DMTs in a wide MS population (MS centers involved in the study cover a population of approximately 5000 MS patients); second, we were able to observe the use of DMTs also in frail populations, including pregnant women, for which concern still exist in the choice to treat MS. Indeed, many pregnant women shared specific concerns about taking DMTs and therefore for this population there is a significant need to provide advice and guidance concerning their proper use in pregnancy and postpartum phase. Third, besides well-known limitations of post-marketing surveillance activities, neurologists who decided to participate have been trained to reporting ADRs through the ICSR established by the AIFA and therefore lesser errors compared to the pharmacovigilance activities performed in routine clinical practice were expected.

## 6. Conclusions

The development of new and effective DMTs for the treatment of multiple sclerosis has totally changed the therapeutic scenario both for neurologists and patients. Since most of these drugs were recently approved for the use in clinical practice, a strict monitoring of their safety profile is strongly recommended, especially if we consider that sometimes these medications can be associated to very serious ADRs. 

With the aim to improve the knowledge on safety aspects of DMTs used in the treatment of MS, we started in September 2018 the FASM pharmacovigilance project, which is dedicated to the monitoring of IFN beta-1a/1b, pegIFN beta-1a, glatiramer acetate, teriflunomide, fingolimod, cladribine, dimethyl fumarate, alemtuzumab, ocrelizumab and natalizumab in the Campania Region. The study is currently carried out across ten hospitals of the Campania Region. During the first 16 months of the FAMS project, 272 ICSRs were collected. Overall, ADRs occur in adult patients and in a slightly higher percentage of female. No substantial differences were found in term of ADRs distribution by seriousness, except for some DMTs, and the majority of reported ADRs showed a favorable outcome. The most commonly reported suspected DMTs were dimethyl fumarate, fingolimod or IFN beta 1a. These medications were commonly associated to the occurrence of ADRs related hematological, gastrointestinal, general, infective or cancer disorders. 

We believe that our results, together with those that will be presented at the end of the study, may bring new knowledge concerning the safety profile of DMTs and their proper use. This will provide the opportunity to draw new recommendations both for neurologists and patients.

## Figures and Tables

**Figure 1 pharmaceuticals-13-00466-f001:**
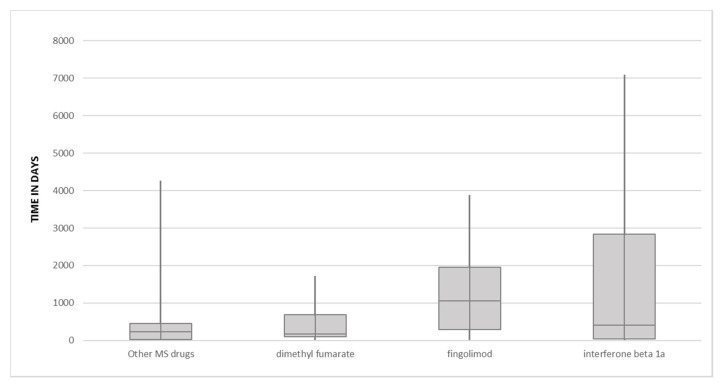
Box plot of median Time to Event in days for dymethil fumarate, fingolimod, interferon beta 1a and other multiple sclerosis (MS) drugs.

**Table 1 pharmaceuticals-13-00466-t001:** Main characteristics of Individual Case Safety Reports (ICSRs) sent to the Rete Nazionale di Farmacovigilanza (RNF) during the first 16 months of the Farmacovigilanza Attiva Sclerosi Multipla (FASM) project.

	All ICSRs (*n* = 272)	ALE (*n* = 11)	CLA (*n* = 7)	DMF (*n* = 41)	FIN (*n* = 62)	GLA (*n* = 27)	IFN Beta 1a (*n* = 41)	IFN Beta 1b (*n* = 5)	NAT (*n* = 21)	OCR (*n* = 25)	PegIFN Beta 1a (*n* = 13)	TER (*n* = 19)
**Median Age** (IQR)	42 (30–50)	39 (27–52)	27 (21–39)	37 (28.5–48.5)	46 (35–53)	43 (37–50.75)	40 (24.5–47.5)	54 (48–63.5)	31 (23–45)	42 (34–59.5)	47 (27.5–58.5)	50 (37–60)
**Sex**												
Female	184 (67.7)	9 (82)	5 (71)	30 (73)	39 (63)	18 (67)	31 (76)	1 (20)	15 (71)	12 (48)	10 (77)	14 (74)
Male	86 (31.6)	2 (18)	2 (29)	11 (27)	23 (37)	9 (33)	9 (22)	4 (80)	6 (29)	13 (52)	2 (15)	5 (26)
NA	2 (0.7)	-	-	-	-	-	1 (2)	-	-	-	1 (8)	-
**Seriousness**												
Not serious	138 (50.7)	3 (27)	3 (43)	25 (61)	26 (42)	14 (52)	24 (59)	4 (80)	8 (38)	14 (56)	8 (62)	9 (47)
Serious	134 (49.3)	8 (73)	4 (57)	16 (39)	36 (58)	13 (48)	17 (41)	1 (20)	13 (62)	11 (44)	5 (38)	10 (53)
**Outcome**												
Favorable	166 (61)	6 (55)	2 (29)	21 (51)	28 (45.2)	21 (77.8)	29 (70.7)	5 (100)	13 (62)	18 (72)	10 (77)	13 (68)
Unfavorable	57 (21)	5 (45)	1 (14)	9 (22)	17 (27.4)	5 (18.5)	8 (19.5)	-	5 (24)	4 (16)	-	3 (16)
NA	49 (18)	-	4 (57)	11 (27)	17 (27.4)	1 (3.7)	4 (9.8)	-	3 (14)	3 (12)	3 (23)	3 (16)
**N. Suspected Drugs Other Than MS Drugs**												
0	269 (99)	11 (100)	7 (100)	41 (100)	62 (100)	26 (96.3)	42 (100)	5 (100)	21 (100)	25 (100)	13 (100)	17 (89)
1	3 (1)	-	-	-	-	1 (3.7)	-	-	-	-	-	2 (11)
**N. Concomitant Drugs**												
0	186 (68.4)	10 (91)	4 (57)	23 (56)	42 (68)	21 (77.8)	30 (73.1)	3 (60)	16 (76)	15 (60)	9 (69.2)	13 (68)
1	40 (14.7)	-	-	11 (27)	10 (16)	4 (14.8)	4 (9.8)	-	4 (19)	3 (12)	2 (15.4)	2 (11)
2	20 (7.4)	1 (9)	2 (29)	3 (7.3)	5 (8)	1 (3.7)	2 (4.9)	1 (20)	1 (5)	-	2 (15.4)	2 (11)
3	9 (3.3)	-	-	1 (2.4)	1 (1.6)	-	2 (4.9)	1 (20)	-	3 (12)	-	1 (5)
4	9 (3.3)	-	-	3 (7.3)	1 (1.6)	-	3 (7.3)	-	-	2 (8)	-	-
≥5	8 (2.9)	-	1 (14)	-	3 (4.8)	1 (3.7)	-	-	-	2 (8)	-	1 (5)

ALE: alemtuzumab; CLA: cladribine; DMF: dimethyl fumarate; FIN: fingolimod; GLA: glatiramer acetate; IFN: interferon; NAT: natalizumab; OCR: ocrelizumab; TER: teriflunomide.

**Table 2 pharmaceuticals-13-00466-t002:** Serious Individual Case Safety Reports (ICSRs) that reported ADRs resulting in death or that were life threating.

Case No.	Age	Sex	Outcome	PT	Suspected Drug 1	Suspected Drug 2	Concomitant Medications	Causality Assessment
1	0	M	Death	Preterm birth	Natalizumab	-	-	Possible
2	0	F	Death	Preterm birth	Glatiramer acetate	Methylprednisolone	-	Possible
3	53	M	Death	Respiratory distress	Ocrelizumab	-	-	Possible
4	42	F	Unchanged	Bacterial meningitis, bacterial pneumonitis	Alemtuzumab	-	-	Possible
5	68	M	Improved	Generalized tonic–clonic seizure, autoimmune encephalitis	Alemtuzumab	-	-	Possible
6	48	M	Unchanged	Limbic encephalitis	Fingolimod	-	Tamsulosin	Possible

**Table 3 pharmaceuticals-13-00466-t003:** Distribution of Adverse Drug Reactions related to dimethyl fumarate, fingolimod and interferon (IFN) beta 1a by System Organ Classes and Preferred Terms.

System Organ Class	DMF (*n* = 53; 100%)	FIN (*n* = 80; 100%)	IFN Beta 1a (*n* = 54; 100%)
**Blood and Lymphatic System Disorders n. (%)**	**17 (32.1)**	**12 (15)**	**11 (20)**
*Anemia*	-	-	1
*Leukocytosis*	1	-	-
*Leucopenia*	1	4	5
*Lymphocytopenia*	12	8	1
*Neutrophilia*	1	-	-
*Neutropenia*	-	-	3
*Thrombocytopenia*	2	-	1
**General Disorders and Administration site Conditions n. (%)**	**1 (1.9)**	**3 (3.75)**	**24 (44)**
*Flu-like symptoms*	1	-	16
*Asthenia*	-	1	1
*Cyst*	-	1	-
*Wheezing*	-	1	-
*Pyrexia*	-	-	2
*Loss of response*			1
*Injection site reaction*	-	-	4
**Infections and Infestations n. (%)**	**4 (7.5)**	**15 (18.75)**	**2 (4)**
*Sepsis*	1	-	-
*Spondylodiscitis*	1	1	-
*Rhinitis*	1	-	-
*Herpes zoster infection*	1	5	-
*Bronchitis*	-	1	-
*Cystitis*	-	4	-
*Encephalitis*	-	1	-
*Molluscum contagious*	-	2	-
*Pneumonitis*		1	-
*Broncho-pneumonitis*	-	-	1
*Cutaneous abscess*	-	-	1
**Neoplasms Benign, Malignant and Unspecified n. (%)**	**1 (1.9)**	**17 (21.25)**	**3 (6)**
*Lymphoma*	1	-	-
*Lung cancer*	-	1	-
*Breast cancer*	-	3	-
*Ovarian cancer*	-	1	-
*Thyroid cancer*	-	1	-
*Epithelioma*	-	1	-
*Melanoma*	-	3	1
*Metastatic gallbladder cancer*	-	1	-
*Myomas*	-	1	-
*Mouth cancer*	-	1	-
*Melanocytic nevus*	-	2	-
*Dysplastic nevus*	-	1	-
*Anal warts*	-	1	-
*Colorectal cancer*	-	-	1
*Nodular fasciitis*	-	-	1
**Investigations n. (%)**	**1 (1.9)**	**12 (15)**	**3 (6)**
*Increase in transaminases*	1	1	3
*ALT elevation*	-	2	-
*GGT elevation*	-	6	-
*Cholesterol elevation*	-	1	-
*GPT elevation*	-	2	-
**Gastrointestinal Disorders n. (%)**	**11 (20.8)**	**2 (2.5)**	**1 (2)**
*Heartburn*	7	-	-
*Dysphagia*	1	-	-
*Diarrhea*	2	-	-
*Unspecified gastrointestinal disorder*	1	-	-
*Pancreatic insufficiency*		1	-
*Nausea*		1	-
*Oral lesion*	-	-	1
**Skin and Subcutaneous Tissue Disorders n. (%)**	**6 (11.3)**	**1 (1.25)**	**5 (9)**
*Eczema*	2	-	-
*Skin eruption*	2	-	-
*Facial redness*	2	-	-
*Skin lesion*	-	1	-
*Sweating*	-	-	1
*Measles rash*	-	-	1
*Hair thinning*	-	-	1
*Skin induration*	-	-	1
*Skin pigmentation*	-	-	1
**Other SOCs n. (%)**	**12 (22.6)**	**18 (22.5)**	**5 (9)**
*Congenital, familial and genetic disorders*	**1**	**-**	**-**
*Hepatobiliary disorders*	**1**	**3**	**2**
*Injury, poisoning and procedural complications*	**2**	**2**	**-**
*Renal and urinary disorders*	**1**	**-**	**-**
*Vascular disorders*	**3**	**-**	**-**
*Respiratory, thoracic and mediastinal disorders*	**4**	**2**	**-**
*Cardiac disorders*	**-**	**2**	**-**
*Eye disorders*	**-**	**3**	**-**
*Nervous system disorders*	**-**	**6**	**1**
*Endocrine disorders*	**-**	**-**	**1**
*Pregnancy, puerperium and perinatal conditions*	**-**	**-**	**1**

n.: The number; DMF: Dimethyl fumarate; FIN: Fingolimod; IFN: interferon; TTE: Time to Event.

**Table 4 pharmaceuticals-13-00466-t004:** Median (IQR) and range of Time to Event (days) for dimethyl fumarate, fingolimod and IFN beta 1a and most common reported System Organ Class.

Median and Range of TTE by SOCs and Suspected Drugs	Dimethyl Fumarate	Fingolimod	IFN Beta 1a
**Blood and Lymphatic System Disorders** Median (IQR) Range	**224** (145–621.5) 145–1629	**1227** (500–1529.75) 74–2575	**1518** (223–2860) 68–3061
**General Disorders and Administration Site Conditions** Median (IQR) Range	**-**	**70** (0–679) 0–679	**188** (0.25–4900) 0–706
**Infections and Infestations** Median (IQR) Range	**943** (139.75–1598.75) 0–1689	**1486.5** (769.25–2609.75) 212–3585	-
**Neoplasms Benign, Malignant and Unspecified** Median (IQR) Range	-	**1589.5** (1040.75–2366) 464–2745	**3681** (2760–3697) 2760–3697
**Investigations** Median (IQR) Range	**-**	**1396** (116.75–2197.75) 64–2474	-
**Gastrointestinal Disorders** Median (IQR) Range	**92** (33–646) 7–1460	-	-
**Skin and Subcutaneous Tissue Disorders** Median (IQR) Range	**679** (145.25–1091) 122–1430	-	**72** (0–583.5) 0–979

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
