# Peer review of "Preliminary Results of the FASM Study, an On-Going Italian Active Pharmacovigilance Project"

_pharmaceuticals, 2020, doi:10.3390/ph13120466_

Round 1

Reviewer 1 Report

This was a really interesting paper and of great value to MS clinicians in particular, but also the wider clinical / patient safety community. 

There is a lot of data but it was easy to read and interpret - you've done a good job of making so much information clear to the reader!

A couple of minor points:

Minor clarification of wording on line 104-108 - re Naranjo algorithm. Rather than "resulted in possible" it would be clearer to say "determined causality as possible" or similar. Also Table 1 "suspected drugs other that MS drugs" should read "other than".

Line 177-180 where you talk about the higher rates in women being unsurprising it would he helpful to add some numbers for context (your approx. 2:1 ratio seems to align with Tullman's study).

In Section 4.1 (and 5) - do you have any information about the level of participation in the scheme? I am conscious that (as you say yourselves) under-reporting of ADRs is a major limitation of much pharmacovigilance research and it would be good to present brief information IF appropriate (ethically) and available, in order to get a sense of the scale / context in which to view the results e.g. reports came from all 10 hospitals or were made by 13 of the 18 neurologists working in the area or that 65% of neurologists in the 10 hospitals received the training to report. (I wasn't sure if the anonymity of the reporting related to just patient anonymity or reporter anonymity too). If not available then obviously ignore this comment!

Author Response

Dear reviewer, first of all we’d like to thank you for having reviewed our manuscript. We took into account all your suggestions and considerations and we hope that now the manuscript could be suitable for the publication.

Regarding to the results of Naranjo algorithm, we changed according to your suggestion in "determined causality as possible" (please see lines 111 and 113). We corrected the typo in Table 1 in "other than".

Regarding to the higher rates of ADRs in women, as you properly suggested we added women to men ration both for MS and ADRs prevalence (please see lines 188-190)

Lastly, regarding to the level of participation in this pharmacovigilance study we have specified in the paragraph 4.1 (lines 315-317) that for each participating center a neurologist plus one/two collaborators collected data on MS drugs-induced ADRs. Of course, we are aware that the under-reporting phenomena could be found in our study even though it is an active pharmacovigilance project. But, at least, we are sure that for each center one neurologist actively participated in the study. So thank for this suggestions!

Reviewer 2 Report

Re-pharmacy MS

General comment:

This is a preliminary report on an ongoing pharmacovigilance study on side effects of drugs for MS. Within 16 months a total of 272 individual case safety reports were communicated. Unfortunately, the ADRs are only listed and not rated. It appears that the drug authority only lists the reports, but not evaluates them and acts accordingly. Therefore, the data are not very helpful and the manuscript is weak, particularly since the preliminary data are incomplete.

Specific comments:

Line 61 and 191: What is a biotech drug in this context?

glatiramer acetate is a synthetic polypeptide,

fingolimod is a synthetically produced Sphingosin-1-phosphat- analogue,

teriflunomide is a synthetically produced inhibitor of dihydroorotate dehydrogenase

dimethyl fumarate is a synthetically produced esther of fumaric acid,

cladribine is a synthetically produced anticancer drug, and

natalizumab, alemtuzumab, and ocrelizumab are monoclonal antibodies and therefore real biological agents.

Line 62: glatiramer acetate was authorised by the EMA in 2001 or 2004 and not in 2005

Line 84: The Individual Case Safety Reports (ICSRs) are reported to a national drug authority. Why  the authority does not  acts?

Line 94: A higher percentage of women compared to men was reported. This is not surprising since the disease occurs to a higher proportion in women.

Line 96: The outcome was reported as favorable in 61% of ICSRs and as unfavorable in 21% of ICSRs. This is in total 82%, What happened with the remaining 18%?

Line 231: not serious with dimethyl fumarate, glatiramer acetate, IFN beta1a and beta1b, ocrelizumab and pegIFN beta 1a;

Line 358: Study strengths and limitations. The first para is a general comment the limitations of the spontaneous reporting of (suspected) ADR to pharmacovigilance authorities. However, the limitation of the study should be reported (preliminary data of an ongoing study, lack of information on the analysis of the data received, transfer to Eurovigliance).

Author Response

Dear reviewer, first of all we’d like to thank you for having reviewed our manuscript. We took into account all your suggestions and considerations and we hope that now the manuscript could be suitable for the publication.

With regard to ADRs that were not listed, we are aware that this is a limitation of our manuscript. But, considering that the results that we presented in this paper were preliminary, we decided to provide data on ADR only for cases that resulted in patients’ death and ADRs related to the three drugs most commonly reported as suspected. At the end of the study, we will present in detail all ADR.

According to your precious suggestions we have replaced the term Biotech with the term Biological and the term most with some (please see lines 63 and 202).

We corrected the authorization date for glatiramer acetate (please see line 64)

We have specified the national drug authority and the national database at lines 90-92

Regarding to the higher percentage of ADR in women compared to men, we have better discussed it by adding data on MS and ADR prevalence in women and men (please see lines 188-190)

With regard to the outcome, we have specified also in the text that in 18% the outcome was not reported (please see lines 102-103); the lack of data, such as that related to the outcome, in each ICSR represents one of the limitation of our study.

Lastly, we have improved the paragraph concerning limitations (please see lines 386, 387, 390,391)

Reviewer 3 Report

Overall, the study is well-conceived, the presentation is good, methods are well-illustrated making the study reproduciple and findings are of some interest.

However, there is a limitation that limits the quality of the section Introduction, which should be addressed to improve the manuscript. When citing the association betweem multiple Sclerosis and Vitamin D, a mention should be done to seminal works which are not reported within the reference section. 

Please, consider citing:

1 PMID: 31142227 

2 PMID: 31284484

3 PMID: 31330127 

4 PMID: 32542086

5 PMID: 28834557

6 PMID: 33159645 

After addressing this issue by adding the mention to these papers within the body text or at least within the reference section, the paper according to this Reviewer could be suitable for publication.

Author Response

Dear reviewer, thank you for your precious suggestions. We have cited in the introduction 4 of the articles you have suggested and we briefly reported that there are still conflicting data on the association between vitamin D and MS (please see lines 54,55)

Reviewer 4 Report

This is a clearly written and extensively referenced manuscript characterized by a wealth of information on DMTs in multiple sclerosis.

Author Response

Dear reviewer, thank you very much for having reviewed our manuscript.

Round 2

Reviewer 2 Report

The specific questions have been answered and the limitations of a preliminary report also have been addressed.

If the JOURNAL accepts preliminary material it can be published as it stands.